# Sarcopenia and Tuberculosis: Is There Any Connection?

**DOI:** 10.3390/jpm13071102

**Published:** 2023-07-06

**Authors:** Nikolaos D. Karakousis, Konstantinos I. Gourgoulianis, Ourania S. Kotsiou

**Affiliations:** 1Department of Respiratory Medicine, Faculty of Medicine, University of Thessaly, Biopolis, 41110 Larissa, Greece; 2Laboratory of Human Pathophysiology, Faculty of Nursing, University of Thessaly, Gaiopolis, 41500 Larissa, Greece

**Keywords:** sarcopenia, tuberculosis, low muscle mass, TB, infection

## Abstract

Background: Tuberculosis (TB) infection is a life-threatening infection caused by certain bacteria belonging to the Mycobacterium tuberculosis complex. More than 10 million subjects are newly sick from this infection every year globally. At the same time, TB is quite prevalent among subjects who come from lower socioeconomic layers of general population, and marginalized sections and areas. Sarcopenia is a muscle disease that derives from adverse muscle alterations and is related to the loss of muscle strength and mass. It is a major medical issue due to its increased adverse outcomes including falls, functional decline, frailty, hospitalizations, increased mortality, and healthcare costs. Methods: This study examined the potential interplay between the TB infection and sarcopenia through conducting a non-systematic review of the current literature. Results: It has been recorded that the prevalence of sarcopenia among TB survivors is high, whilst the danger of TB among the elderly increases with sarcopenia and physical inactivity. Nevertheless, sufficient protein and total energy intake are associated with a low risk of sarcopenia in TB survivors. Conclusions: Further studies are needed to validate these findings and shed more light on the upcoming different aspects of this intriguing association.

## 1. Introduction

Tuberculosis (TB) infection is still one of the most frequent causes of death among adult subjects globally, in the area of infectious diseases, with more than 10 million subjects becoming sick with this infection every year [1]. It is well established that TB is quite prevalent among subjects who come from lower socioeconomic layers of general population, and marginalized sections and areas all over the world [2].

Sarcopenia is a muscle disease which derives from adverse muscle alterations that can occur during someone’s life, being quite frequent among subjects of older age; however, it can also be present in subjects of a younger age, along with specific pathological medical issues [3,4,5]. It is a clinical condition associated with loss of muscle mass and strength, and even though this might lead to adverse events, a single and specific diagnostic criterion has not yet been validated [6]. The existence of sarcopenia is strongly associated with poorer health outcomes throughout an individual’s life [7].

In this non-systematic review, we tried to investigate the potential and upcoming interplay between the clinical condition of sarcopenia and TB infection. It is already reported that muscle weakness might be a part of the wasting syndrome associated with TB [8]. Consequently, it is essential that we further investigate the association between these two entities, and further reveal their role and impact on the individuals’ health course when these two life-threatening conditions coexist.

The term sarcopenia has Hellenic roots, and derives from the word ‘sarx’ which describes the flesh, and ‘penia’ which describes and concerns loss. It is closely associated with significant alterations in body composition and related body functions [9,10]. The pathogenesis of this clinical condition relies on the balance between positive and negative regulators that concern muscle growth [11].

Sarcopenia is related to the loss of muscle strength and mass in older individuals, but it can also be present among younger individuals with various other comorbidities [4,12,13]. In recent years, this clinical condition has become a significant issue concerning both research and public policy debate, and the main reasons for this outcome are increased adverse outcomes including falls, functional decline, frailty, and hospitalizations, along with increased morbidity, mortality, and healthcare costs [12,14].

It is recorded in the current literature that in the year 2010, the European Working Group on Sarcopenia in Older People (EWGSOP) published a definition concerning sarcopenia that aimed to point out the latest advances in diagnosing and caring for individuals living with sarcopenia [3]. Nevertheless, in early 2018, the Working Group EWGSOP2 met again to update the original definition and demonstrate scientific and clinical evidence that has been collated over the last decade [3,6]. The updated consensus on sarcopenia (EWGSOP2) has the following key points: (a) it focuses especially on low muscle strength as a key feature of sarcopenia, utilizing the detection of low muscle quantity and quality to confirm the diagnosis of sarcopenia, and identifies poor physical performance as indicative of a severe form of sarcopenia; (b) it updates the clinical algorithm to be used for sarcopenia case-finding, diagnosis and confirmation, along with severity determination; and (c) it provides clear cut-off points for measurements of variables that identify and define sarcopenia [3].

Consequently, the three following criteria: (i) low muscle strength, (ii) low muscle quantity or quality, and (iii) low physical performance operationally define probable sarcopenia if criterion (i) is met, a confirmed diagnosis of sarcopenia if there is additional documentation of criterion (ii), and severe sarcopenia if criteria (i), (ii), and (iii) are all met [3]. Moreover, specific tests and tools, such as measuring grip strength and the chair stand test, along with questionnaires such as the SARC-F questionnaire, might be used in order to further identify sarcopenia in practice and in research [3,15,16]. Muscle quantity or mass can be reported as appendicular skeletal muscle mass (ASM), total body skeletal muscle mass (SMM), or as the muscle cross-sectional area of specific muscle groups or human body locations [3]. The gold standards for non-invasive assessment of muscle quantity/mass are magnetic resonance imaging (MRI) and computed tomography (CT), whilst dual-energy X-ray absorptiometry (DXA) is a more widely available mean to determine muscle quantity (appendicular skeletal muscle mass or total body lean tissue mass) in a non-invasive manner [3,17,18]. Finally, Bioelectrical Impedance Analysis (BIA) has been examined to estimate total or ASM; it does not measure muscle mass directly, but estimates the muscle mass based on whole-body electrical conductivity [3,19]. Nevertheless, various tools and other tests are also being investigated [15].

Concerning potential interventions to counteract this clinical condition, many means have been suggested, focusing mainly on training and exercise programs and nutrition, but currently there are no specific guidelines [20,21,22]. It has been recorded that dietary interventions that involve protein intake ameliorated functional and/or strength outcomes in a few trials, while other dietary interventions were less effective [23]. On the other hand, exercise and training programs, specifically resistance training programs, have already been characterized as the most promising way to increase muscle mass and strength, especially among older people [24]. It has been recorded that a regular exercise program, for example, exercising three times/week and including resistance and endurance exercise training, can significantly positively impact sarcopenic muscle, improving muscle strength, mass, and function [25].

TB, as an infection, has accompanied the history of human development from the Stone Age until now [26]. More specifically, the history of TB has been traced back to the Stone Age and Paleolithic period, approximately 3.3 million years ago; it reached epidemic levels in North America and Europe in the eighteenth and nineteenth centuries [26]. TB infection is still one of the leading causes of death from infectious diseases globally [1]. It is well established that this kind of infection is caused by certain bacteria that belong to the Mycobacterium tuberculosis complex, which is one of the oldest diseases known to have a detrimental impact on human health and homeostasis [2,27].

Concerning diagnosis, if a physician’s differential diagnosis includes TB, the main tests available for isolating the pathogen in both tissue and sputum samples are culture, which remains the gold standard, microscopy, and nucleic acid amplification tests [28]. Recent advances in molecular diagnostic procedures including loop-mediated isothermal amplification (LAMP), GeneXpert, MTBDRplus, line probe assay (LPA), and whole genome sequencing (WGS) have been utilized to diagnose and characterize TB infection, and these methods may further identify Mycobacterium tuberculosis (MTB) and mutation(s) associated with routinely used anti-TB drugs [29]. Additionally, imaging is significant concerning the diagnosis and follow-up of TB subjects [28,30]. Concerning primary pulmonary TB, chest radiography is still the mainstay for diagnosing parenchymal disease, whilst CT is more sensitive to detecting lymphadenopathy [30]. In post-primary pulmonary TB, CT can reveal early bronchogenic spread. In addition, concerning the characterization of the infection as active or not, CT is more sensitive than radiography, while (18) F-fluorodeoxyglucose positron emission tomography/CT [(18) F-FDG PET/CT] has provided promising outcomes [30]. MRI is preferable for diagnosing and evaluating tuberculous spondylitis, whilst (18) F-FDG PET demonstrates superior image resolution compared to single-photon-emitting tracers [30]. Finally, MRI is considered superior to CT for detecting and evaluating the central nervous system located in TB [30].

Treatment usually combines specific anti-TB agents such as isoniazid, rifampicin, ethambutol, and pyrazinamide. At the same time, it has already been recorded that liver damage is one of the more common adverse effects of this therapy, arising in approximately 2.4% of TB subjects [28,31]; meanwhile, multidrug-resistant (MDR) TB can also be a significant issue for both the patient and attending physician [28,32,33]. It is shown in the current literature that India has the highest incidence of new and MDR TB cases in the world [34]. Concerning prevention, Bacille Calmette–Guerin (BCG) is the vaccine that is commonly available for deterring this infection. Nevertheless, this vaccine may offer an appropriate defense against serious forms of TB in childhood, but its protective effect is reduced with age [34,35]. New tactics are being used to develop more effective and powerful vaccines, such as mucosal- and epitope-based vaccines [34].

The fact that it is already recorded in the current literature that muscle weakness is a part of the wasting syndrome associated with TB [8] provides the appropriate incentive to investigate it further, and dive into the potential association between sarcopenia and TB infection by producing this non-systematic review article. In addition, these two entities might share common causes that could lead to their occurrence. One significant example is the socioeconomic disadvantages some subjects may sustain that could fuel both the clinical condition of sarcopenia and a greater risk of infections such as TB, due to poor hygiene [36,37]. At the same time, the lack of a proper health education strategy could hinder our progress in both sarcopenia and TB’s evolution and prognosis [38,39].

## 2. Materials and Methods

We conducted an electronic search in the databases of PubMed, Google Scholar and EMBASE from 29 July 1989 till June 2023 using the following combinations of specific keywords: “sarcopenia” OR “low muscle mass” AND “tuberculosis”. Only original articles which written in English language were included in our non-systematic review article. Additionally, all the references of the included studies were thoroughly examined. Studies concerning animals were excluded from this review study. The literature review’s organization is summarized in the flowchart diagram (Figure 1).

## 3. Results

The main purpose of this study was to reveal the existence of any potential and upcoming association between sarcopenia and TB infection, as depicted by the current literature, based on the potentially life-threatening profile of each of these two medical conditions. The results of our non-systematic review are presented in Table 1.

Shin et al., tried to assess the prevalence of sarcopenia and its association with total energy and protein intake among Korean TB survivors, because TB causes undernutrition and it has a lengthy recovery period after treatment implementation; it is also accompanied by adverse outcomes, among them sarcopenia and low muscle mass [40]. The definition of TB survivors includes individuals with a self-reported previous history of physician-diagnosed TB, or those with a healed TB lesion upon chest X-ray (CXR) [40]. The authors conducted a national, cross-sectional, population-based study including 9.203 participants aged ≥ 40 years, and 962 (9.7%) were TB survivors [40]. They utilized three definitions for sarcopenia-appendicular skeletal muscle mass (ASM, kg) divided by body mass index (BMI, kg/m^2^), weight (kg), or height squared (m^2^), whilst the daily total energy and protein intake were estimated using a 24 h recall method [40]. A multiple logistic regression was utilized to assess the relationship between dietary protein/total energy intake and sarcopenia among TB survivors [40]. Their results included that the prevalence of sarcopenia was 11.2%, 10.7%, and 24.3% among TB survivors with sarcopenia defined by ASM divided by BMI, weight, and height squared, respectively [40]. Additionally, they concluded that the prevalence of sarcopenia among TB survivors was higher than among those without TB. At the same time, sufficient protein and total energy intakes were related to a lower risk of sarcopenia in TB survivors [40].

In another study Choi et al., tried to evaluate and demonstrate if there is a potential risk of sarcopenia and osteoporosis among Korean male TB survivors [41]. Among 3228 male individuals aged 50 years or older, 529 (16.4%) were TB survivors. Of these, 98 (3.0%) only had a history of TB, 245 (7.6%) only had radiographic evidence of TB and specifically on CXR, and 186 (5.8%) had both a history and CXR evidence of TB [41]. In this population-based study, they concluded that TB survivors with both a medical history and TB scars on CXR had an increased risk of sarcopenia (odds ratio [OR] 3.44, 95% confidence interval [CI] 1.79–6.68) and osteoporosis (OR 1.75, 95% CI 1.04–2.95) after adjusting for age, height, alcohol, smoking, physical activity, parathyroid hormone level, serum 25-hydroxyvitamin D, education, and fat mass index [41]. In addition, having TB scars on CXR without a medical history of TB was an independent risk factor for sarcopenia (OR 2.05, 95% CI 1.05–4.00), even though it was not a risk factor of osteoporosis [41]. Consequently, low bone mineral density and sarcopenia might be prevalent in pulmonary TB survivors with TB scars on CXR. In contrast, a medical history of TB with TB scars on CXR is an independent risk factor for osteoporosis and sarcopenia [41].

Nevertheless, in another study, Yoo et al. examined whether parameters such as sarcopenia, physical activity, and anemia are related to an increased risk of TB, specifically among the older population [42]. Their population-based study included 1,245,640 individuals of 66 years old who participated in the National Screening Program for Transitional Ages for Koreans from 2009 to 2014, with a median follow-up duration of 6.4 years [42]. At baseline, they evaluated common health issues in the older population, among them sarcopenia and anemia, whilst the individuals’ performance in the timed up-and-go (TUG) test was utilized in order to predict sarcopenia [42]. The TUG test was conducted on the examination day, during which the individuals were observed and timed while rising from an armchair, walking three meters, turning, walking back, and sitting down again [42]. In addition, the individuals were instructed to wear regular footwear and use their usual walking aid [42]. For their analyses, the TUG test results were categorized into <10 s, 10–15 s, and ≥15 s, after considering the distribution of the TUG outcomes [42]. Among their results, it was quite intriguing that compared with those who had normal TUG times, subjects with slow TUG times (≥15 s) had a significantly increased risk of TB [adjusted hazard ratio (aHR): 1.19, 95% CI: 1.07–1.33], while both irregular (aHR: 0.88, 95% CI: 0.83–0.93) and regular (aHR: 0.84, 95% CI: 0.78–0.92) physical activity reduced the risk of TB [42]. As a result, the authors concluded that the risk of TB among the elderly is relatively increased with sarcopenia and physical inactivity [42].

The study of Tanaka et al., based on the fact that skeletal muscle size is a quite significant parameter and is considered a predictor of prognosis in patients with respiratory diseases, including mycobacterium avium complex lung infection, investigated the linkage between erector spinae muscle (ESM) size and in-hospital mortality among subjects with pulmonary TB [43]. Their retrospective study included 258 consecutive subjects aged over 65 years old who were admitted to the hospital for pulmonary TB, as bacteriologically confirmed, and all underwent a chest CT scan upon admission [43]. Additionally, the measurement of the cross-sectional area of the ESM (ESMcsa) was conducted at the lower margin of the 12th thoracic vertebra on a single-slice CT scan image, and was adjusted according to body surface area (BSA); 71 (28%) subjects died during hospitalization overall [43]. In the non-survivor group, high incidences of respiratory failure and comorbidities were recorded, as well as lower hemoglobin and albumin levels, performance status scores, and ESMcsa/BSA. The multivariate analysis demonstrated that low performance status scores and albumin and hemoglobin levels, but not body mass index and ESMcsa/BSA, could independently predict in-hospital mortality after adjusting for age and comorbidities [43]. As a result, ESM size was not associated with in-hospital mortality in patients with pulmonary TB [43].

Finally, in another study conducted by Villamor et al., the authors aimed to examine the impact of HIV coinfection, socioeconomic status (SES), and the severity of TB on the body composition and anthropometric status of adult subjects with pulmonary TB [44]. In their cross-sectional study, they included 2231 subjects, where 69% were men, 31% were women, and all of them had been diagnosed with pulmonary TB prior to the initiation of anti-TB treatment; they compared the distribution of anthropometric features including body mass index (BMI), tricep skin fold (TSF), mid-upper arm circumference (MUAC), and arm muscle circumference (AMC) with HIV status, SES characteristics, and indicators of TB severity (e.g., bacillary density in sputum and the Karnofsky performance score) [44]. Additionally, similar comparisons were conducted with body composition variables from a bioelectrical impedance analysis and albumin concentrations of a subsample of 731 individuals [44]. Interestingly, upon multivariate analysis, HIV infection was significantly related to lower MUAC and AMC in both women and men, but not associated with BMI or TSF [44]. Compared to HIV-uninfected women, those who were HIV-infected had a lower body cell mass (BCM) (adjusted difference = −0.85 kg, *p* = 0.04), intracellular water level (−0.68 L, *p* = 0.04), and phase angle (−0.52, *p* = 0.02), while independently of HIV infection, both BMI and MUAC were positively associated with SES indicators and the Karnofsky performance score, and inversely associated with bacillary density [44]. Consequently, Villamor et al. concluded that HIV infection is related to low lean body mass indicators in adult subjects with TB [44].

**Table 1 jpm-13-01102-t001:** Association between sarcopenia and tuberculosis infection.

Authors/[Reference]	Study Design	Study Population	Main Results	Muscle Mass Evaluation
Shin et al./[40]	Cross-sectional, population-based study	9203 participants aged ≥ 40 years	The prevalence of sarcopenia among TB survivors was higher than among those without TB. Sufficient protein and total energy intake were associated with a lower risk of sarcopenia in TB survivors	ASM (kg) divided by BMI (kg/m^2^), weight (kg), or height squared (m^2^)
Choi et al./[41]	Population-based study	529 (16.4%) TB survivors	Low BMD and sarcopenia might be prevalent in pulmonary TB survivors with TB scars on CXR, while a medical history of TB with TB scars on CXR was an independent risk factor for osteoporosis and sarcopenia	ASM, ASMI
Yoo et al./[42]	Population-based study	1,245,640 individuals 66-year-old	The risk of TB among the elderly was increased with sarcopenia and physical inactivity	TUG test
Tanaka et al./[43]	Retrospective cohort study	258 subjects aged over 65 years old	ESM size was not associated with in-hospital mortality in patients with pulmonary TB	CT scan—ESMcsa/BSA
Villamor et al./[44]	Cross-sectional study	2231 subjects (69% men-31% women)	HIV infection was related to indicators of low lean body mass in adult TB subjects	TSF, MUAC, AMC, BIA

Abbreviations: TB: tuberculosis; ASM: appendicular skeletal muscle mass; kg: kilograms; BMI: body mass index; m: meters; BMD: bone mineral density; ASMI: appendicular skeletal mass index by height squared; TUG: timed up-and-go; ESM: erector spinae muscle; CT: computed tomography; ESMcsa: cross-sectional area of the erector spinae muscle; BSA: body surface area; HIV: human immunodeficiency virus; TSF: triceps skin-fold; MUAC: mid-upper arm circumference; AMC: arm muscle circumference; BIA: bioelectrical impedance analysis.

## 4. Discussion

In this non-systematic review, we recorded the existing data concerning the interplay between low muscle mass sarcopenia and TB infection, along with each entity’s impact on the other. These two entities generally share a connection, as was initially hypothesized and triggered our investigation.

Nevertheless, despite the thorough examination that we have conducted, certain limitations should be taken under serious consideration. Firstly, the existing data are limited and scarce, with quite a few studies investigating this interplay. In addition, the number of subjects participating is mainly small, while the follow-up period is short. Moreover, another significant limitation of this study is the fact that it is based mainly on observational studies, and it is already well established that these studies are more prone to bias and cannot be broadly utilized to demonstrate causality.

Overall, a significant result of this non-systematic review is the debate that it creates concerning future perspectives regarding this interplay and these two entities. For example, it would be quite interesting to apply this investigation to a more significant number of TB subjects, in whom we can assess and evaluate muscle mass status, even using a specific index and score that might be specialized to this infection only. Moreover, results derived from a group of TB patients could be further validated using experimental in vitro models, which we could study the muscle tissues of TB subjects along with the isolation of specific biomarkers that could be indicative of the coexistence of these two entities and that could be used as diagnostic or even prognostic tools. Another significant tool could be the use of Mendelian randomization (MR), which is a highly efficient method that allows researchers to identify a causal relationship between phenotypes (diseases) and estimates of causal effects, thus overcoming the limitations of observational studies [45]. To perform such an analysis and to establish a possible connection between sarcopenia and TB, a list of DNA polymorphisms, known as single-nucleotide polymorphisms (SNPs), which are associated with these diseases could be utilized and investigated [46,47]. As a result, in order to further clarify the association between the entities of TB infection and sarcopenia in the future, it may be appropriate to carry out MR using previously known genetic tools.

Concerning therapeutic interventions, another intriguing question and field of upcoming research could be the study of anti-TB pharmaceutical agents and the impact that each agent might have on ameliorating or deteriorating muscle mass health in TB subjects, not only through the type of prescribed medication, but also through combinations of anti-TB agents which might be more beneficial or less harmful to muscle and skeletal muscle status.

Finally, a significant aspect of treating sarcopenia is that of nutritional interventions and the implementation of specific exercise and training programs. It would be remarkable if we could investigate the role of nutrition in treating TB subjects with sarcopenia and thereby discover the exact nutrients that could fortify these subjects and their muscle mass against sarcopenia. The same is true of exercise interventions and the validation of specific training programs that could reverse any adverse outcomes in individuals living with the coexistence of these two entities.

Consequently, it appears that the management of TB subjects living with sarcopenia (and of sarcopenic individuals living with TB infection) concerns not only laboratory scientists and researchers, but also everyday clinical physicians, whose patients present with infectious lung disease such as TB, and nutritionists and fitness instructors, who must cooperate along with the patients in order to provide an optimal approach.

## 5. Conclusions

As has been portrayed in this non-systematic review article, in studying the existing literature thoroughly, we have observed an upcoming and potential association and interplay between low muscle mass sarcopenia and TB infection. Consequently, it seems that the prevalence of sarcopenia among TB survivors is higher; in particular, sarcopenia might be prevalent in pulmonary TB survivors with TB scars on CXR, while TB and HIV coinfection might be related to indicators of low lean body mass in adult TB patients. In addition, the danger of TB infection among older subjects seems to be high in individuals with sarcopenia and physical inactivity. Nevertheless, ESM size seems not to be related to in-hospital mortality in subjects with pulmonary TB; one bright outcome of this review is that we observed that sufficient protein and total energy intake are associated with lower risk of sarcopenia in TB survivors. All these results may benefit everyday physicians who work with TB survivors, and should raise awareness among them that the examination of the muscle and skeletal health conditions should not be neglected in these types of individuals. In conclusion, it is more than imperative that further studies be conducted soon in order to shed more light on this intriguing medical debate.

## Figures and Tables

**Figure 1 jpm-13-01102-f001:**
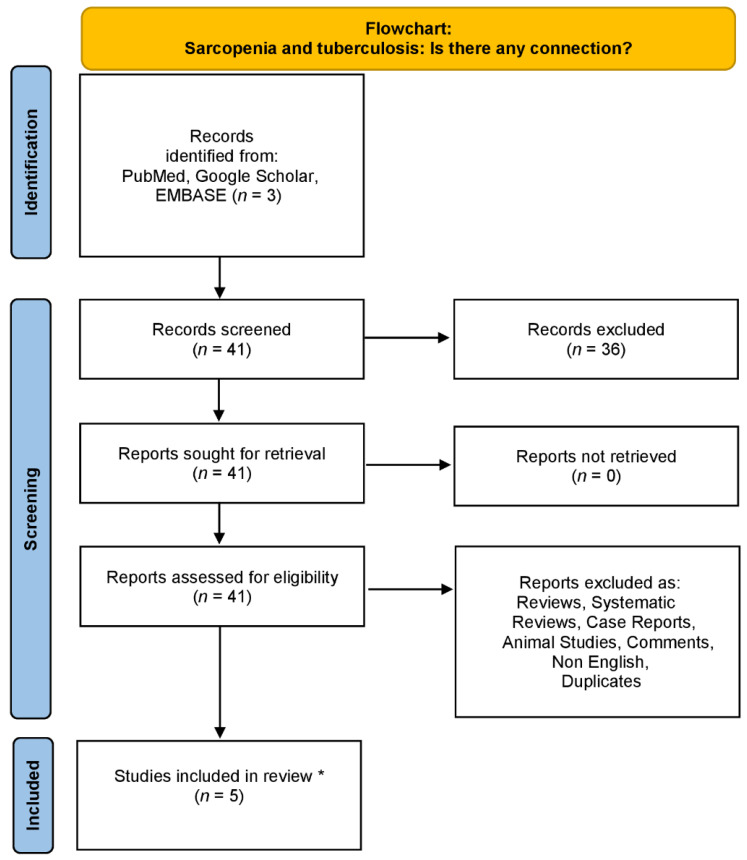
Flowchart diagram demonstrating the literature review strategy (* only original non-animal studies written in English were included).

## Data Availability

Not applicable.

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
