# Peer review of "Sarcopenia and Tuberculosis: Is There Any Connection?"

_jpm, 2023, doi:10.3390/jpm13071102_

Round 1

Reviewer 1 Report

Minor Comments

This research article investigates the potential interplay between the TB infection and sarcopenia conducting a non-systematic review of the current literature. The manuscript is well-written, and the experimental design/data analysis is robust.  I would recommend the following minor comments to the authors.

Point 1: It would be better to do some statistical analysis of the reviewed data and add to the article if possible.

Point 2: Did you used any other specific criteria for the inclusion and exclusion of the data.

Point 3: Minor English correction is required in the revised version.

Point 4: Overall, I could not really fault the experiments or the interpretation.

Good Luck

 Minor editing of the English language required

Author Response

July 2023

Dear Editor of JPM,

We wish to thank you for considering our review article for publication in Journal of Personalized Medicine and for allowing us to resubmit a revised version of our manuscript.  In addition, we would like to thank the reviewers for their insightful comments and suggestions, which have been extremely helpful in order to improve and strengthen our manuscript. We have carefully revised our manuscript, according to the comments made by the reviewers and please find below the changes that we have made, using the green color highlight mode in MS Word, to our initial submission.

RESPONSE TO THE REFEREES

# Reviewer 1

This research article investigates the potential interplay between the TB infection and sarcopenia conducting a non-systematic review of the current literature. The manuscript is well-written, and the experimental design/data analysis is robust. I would recommend the following minor comments to the authors.

Response: Thank you for your encouraging and positive comment.

Point 1: It would be better to do some statistical analysis of the reviewed data and add to the article if possible.

Response: Thank you for your comment. At this moment, this is a non-systematic review article, so we have not applied any statistical method.

Point 2: Did you used any other specific criteria for the inclusion and exclusion of the data.

Response: Thank you for your insightful comment. The main inclusion and exclusion criteria of this non-systematic review are portrayed at the flowchart.  

Point 3: Minor English correction is required in the revised version.

Response: Thank you for your comment. An English correction has been applied to our manuscript.

Point 4: Overall, I could not really fault the experiments or the interpretation.

Response: Thank you for your very kind comment.

Good Luck

Response: We are obliged and thankful.

Looking forward to hearing from you in due course,

Nikolaos D. Karakousis, MD, MSc

Reviewer 2 Report

The main issue with this review is that the authors did not attempt to explain in detail the causal relationship between tuberculosis and sarcopenia. My recommendation to the authors is to list in detail the risk factors for both TB and sarcopenia, and then look for common factors. Then try to explain the causal relationship. For example, a common factor might be educational attainment. A low level of education can lead to the development of both sarcopenia (PMID: 36346726; PMID: 36771461) and tuberculosis. Further, it can be assumed that low educational attainment leads to a low level of hygiene, which in turn leads to an increased risk of infection with tubercle bacillus. Further, infection with a tubercle bacillus leads to sarcopenia. Thus, low educational attainment leads to sarcopenia due to the facts that a person with low education is more likely to become infected, does not have access to high-quality medicine, does not have the opportunity to buy quality food, does not know well the principles of a healthy lifestyle, and so on. Perhaps a figure or table would be appropriate to explain the relationship between tuberculosis and sarcopenia.

In the Discussion, I recommend adding as a perspective the implementation of the Mendelian randomization method, which allows to establish the influence of one factor on another (for example, the effect of tuberculosis on the risk of sarcopenia or the effect of sarcopenia on the risk of tuberculosis or mutual influence on each other). See PMID: 36018815 and PMID: 36123897 for examples. Probably, TB and sarcopenia share many common risk genetic variants.

Author Response

July 2023

Dear Editor of JPM,

We wish to thank you for considering our review article for publication in Journal of Personalized Medicine and for allowing us to resubmit a revised version of our manuscript.  In addition, we would like to thank the reviewers for their insightful comments and suggestions, which have been extremely helpful in order to improve and strengthen our manuscript. We have carefully revised our manuscript, according to the comments made by the reviewers and please find below the changes that we have made, using the green color highlight mode in MS Word, to our initial submission.

RESPONSE TO THE REFEREES

#Reviewer 2

The main issue with this review is that the authors did not attempt to explain in detail the causal relationship between tuberculosis and sarcopenia. My recommendation to the authors is to list in detail the risk factors for both TB and sarcopenia, and then look for common factors. Then try to explain the causal relationship. For example, a common factor might be educational attainment. A low level of education can lead to the development of both sarcopenia (PMID: 36346726; PMID: 36771461) and tuberculosis. Further, it can be assumed that low educational attainment leads to a low level of hygiene, which in turn leads to an increased risk of infection with tubercle bacillus. Further, infection with a tubercle bacillus leads to sarcopenia. Thus, low educational attainment leads to sarcopenia due to the facts that a person with low education is more likely to become infected, does not have access to high-quality medicine, does not have the opportunity to buy quality food, does not know well the principles of a healthy lifestyle, and so on. Perhaps a figure or table would be appropriate to explain the relationship between tuberculosis and sarcopenia.

Response: Thank you for this important comment. In our revised manuscript we have added significant information about common factors that could fuel this kind of interplay between these two clinical entities at the end of our introduction section, in order to strengthen our idea for investigating this association. Socioeconomic factors are mainly underlined, as you correctly mentioned at your significant comment.

In the Discussion, I recommend adding as a perspective the implementation of the Mendelian randomization method, which allows to establish the influence of one factor on another (for example, the effect of tuberculosis on the risk of sarcopenia or the effect of sarcopenia on the risk of tuberculosis or mutual influence on each other). See PMID: 36018815 and PMID: 36123897 for examples. Probably, TB and sarcopenia share many common risk genetic variants.

Response: Thank you for this quite intriguing comment. It seems that there are not specific and validated genetic variants underlined by the current literature that could be related to both these conditions, but this is a brilliant idea that could be the basis concerning further scientific investigation.

Looking forward to hearing from you in due course,

Nikolaos D. Karakousis, MD, MSc

Round 2

Reviewer 2 Report

It seems that the authors did not understand what I wrote earlier. I'll try to explain again. There is a highly efficient method that allows the researchers to identify a causal relationship between phenotypes (diseases). It's called Mendelian randomization. To conduct such an analysis, for example, to establish a connection between tuberculosis and sarcopenia, it is necessary to know the list of DNA polymorphisms (SNPs) associated with these diseases. These lists are already known and mentioned here:
https://journals.lww.com/md-journal/Fulltext/2022/09160/Causal_relationship_between_insomnia_and.62.aspx
https://www.mdpi.com/2072-6643/15/3/758
Therefore, the authors need to write in the Discussion as a perspective that in order to clarify the issue of the relationship between tuberculosis and sarcopenia, it would be appropriate in the future to carry out Mendelian randomization using known genetic tools.

Author Response

July 2023

Dear Editor of JPM,

We wish to thank you for considering our review article for publication in Journal of Personalized Medicine and for allowing us to resubmit a revised version of our manuscript.  In addition, we would like to thank the reviewers for their insightful comments and suggestions, which have been extremely helpful in order to improve and strengthen our manuscript. We have carefully revised our manuscript, according to the comments made by the reviewers and please find below the changes that we have made, using the green color highlight mode in MS Word, to our initial submission.

RESPONSE TO THE REFEREES

#Reviewer 2

It seems that the authors did not understand what I wrote earlier. I'll try to explain again. There is a highly efficient method that allows the researchers to identify a causal relationship between phenotypes (diseases). It's called Mendelian randomization. To conduct such an analysis, for example, to establish a connection between tuberculosis and sarcopenia, it is necessary to know the list of DNA polymorphisms (SNPs) associated with these diseases. These lists are already known and mentioned here:

https://journals.lww.com/md-journal/Fulltext/2022/09160/Causal_relationship_between_insomnia_and.62.aspx

https://www.mdpi.com/2072-6643/15/3/758

Therefore, the authors need to write in the Discussion as a perspective that in order to clarify the issue of the relationship between tuberculosis and sarcopenia, it would be appropriate in the future to carry out Mendelian randomization using known genetic tools.

Response: Thank you for this important comment and the clarification. We are obliged for the opportunity to further improve and assess our manuscript. In our revised manuscript we have added significant information about Mendelian randomization in the section of discussion, whilst we have discussed the importance of SNPs concerning different phenotypes.

Looking forward to hearing from you in due course,

Nikolaos D. Karakousis, MD, MSc

Round 3

Reviewer 2 Report

No additional comments.